

# Rapid authenticity testing of artificially bred green turtles (*Chelonia mydas*) using microsatellite and mitochondrial DNA markers

Ting Zhang[1,*], Liu Lin[1,*], Daniel Gaillard[2], Fang Chen[3], Huaiqing Chen[4], Meimei Li[1], Shannan Wu[1], Zhao Wang[1] and Haitao Shi[1]

[1] Ministry of Education Key Laboratory for Ecology of Tropical Islands, Key Laboratory of Tropical Animal and Plant Ecology of Hainan Province, College of Life Sciences, Hainan Normal University, Haikou, China
[2] School of Science, Technology, and Mathematics, Dalton State College, Dalton GA, USA
[3] National Aquatic Wildlife Conservation Association, Beijing, China
[4] Center for Nature and Society, School of Life Sciences, Peking University, Beijing, China
[*] These authors contributed equally to this work.

Corresponding author
Haitao Shi, haitao-shi@263.net

## ABSTRACT

Sea turtles are threatened by climate change and human activity, and their global populations continue to decline sharply. The Chinese government encourages artificial breeding of sea turtles to reduce the use of wild populations. However, artificial breeding of sea turtles is still fairly difficult, and some facilities may illegally purchase wild turtle eggs and then sell incubated turtles by marketing them as artificially bred turtles, which adds another threat to an already endangered species. Therefore, it is necessary to find a reliable method to distinguish the authenticity of artificially bred individuals. In this study, we investigated a turtle farm in southern China, that contained more than 400 green turtles, which were claimed to have been bred in captivity. Parentage testing of turtles from this farm was successfully conducted using two nuclear microsatellites combined with a mitochondrial D-loop DNA marker. Genetic matching of all 19 adults and randomly selected 16 juvenile turtles revealed that none of the juvenile turtles had a matching parent combination among the adult turtles. Therefore, we speculated that the green turtles in this farm were from the wild and that their origin of birth was mainly the Sulu Sea. The methods and molecular markers used in this study could be a reference for rapid authenticity testing of green turtles in future forensic enforcement and population management.

## INTRODUCTION

Sea turtles are umbrella species of the marine ecosystem and flagship species of marine conservation (*Bouchard & Bjorndal, 2000*; *Hamann et al., 2010*). However, the global population and distribution of the sea turtle have sharply reduced due to long-term threats, such as over-exploitation, illegal trade, marine pollution, and climate change (*Chan et al., 2007*; *Hawkes et al., 2009*; *Lam et al., 2011*). All five sea turtle species distributed in China

have been included in the list of Chinese National Fist-Level Key Protected Wild Animals and the Appendix I of Convention on International Trade in Endangered Species of Wild Fauna and Flora (*CITES, 2019*), indicating that sea turtles are strictly protected and the international trade of these species is prohibited except when the purpose of the import is not commercial.

The fast-growing demand for sea turtle displays from Chinese aquariums and private individuals has led to a large-scale illegal trade of live turtles, seriously threatening the survival of wild sea turtles (*Lin et al., 2021*). The Chinese government issued the "Sea Turtle Conservation Action Plan (2019–2033)" in 2018 and the plan recommended "strengthening research on the technology of artificial breeding of sea turtles and promoting the use of artificial breeding sea turtles for aquarium display, publicity, and education to reduce human demand for sea turtles in the wild" (*Chinese Ministry of Agriculture and Rural Affairs, 2019*). Therefore, the number of institutions for artificial breeding of sea turtles is expected to gradually increase in China. However, artificial breeding of sea turtles is still fairly difficult and only a few successful cases have been reported worldwide, such as the Cayman Islands, UK (*Barbanti et al., 2019*), Miami Seaquarium, FL, USA (*Sizemore, 2002*), Sea Life Park, Hawaii, USA (*Wood & Wood, 1980*), and Ishigaki Island in Japan (*Shimizu et al., 2007*). In China, only the Huidong Turtle National Nature Reserve in Guangdong and Haichang Whale Shark Aquarium in Shandong have successfully bred turtles using wild parent turtles under captive conditions (*Gao, Ye & Chen, 2015*; *Li, 2016*). Due to inefficient breeding methods, some facilities illegally purchased wild turtle eggs from the South China Sea or Southeast Asia, and then sold the incubated turtles to aquariums or private organizations by marketing them as artificially bred turtles (*Shanker & Pilcher, 2003*; *Lam et al., 2011*). It is well-known that turtle farms are usually the biggest purchasers of wild turtles in China (*Shi et al., 2007*), and the origins of many turtles in captivity breeding and trade are false or unclear (*Parham & Shi, 2001*; *Parham et al., 2001*). Therefore, to prevent the illegal use of wild populations, it is necessary to develop reliable methods to distinguish the authenticity of artificially bred sea turtles in forensic enforcement and population management.

DNA analysis methods have been widely used in wildlife forensic enforcement. Among these methods, the nuclear microsatellite gene is widely used in molecular marker-assisted breeding and parentage test of captive animals; it has the advantages of high variability and polymorphism, as well as codominance (*Schlötterer & Pemberton, 1994*). The nuclear microsatellite gene can be stably passed from one generation to the other, that is, half of the genetic material of the offspring is contributed by the father and half by the mother. Therefore, on a specific microsatellite locus, two alleles of the offspring should be found in the genotype of the father or the mother, otherwise, they do not have a parent–offspring relationship (*Huang & Wang, 2004*; *Zi-Tuo & Liu, 2014*; *Zhang et al., 2018*). Mitochondrial sequences have the characteristics of maternal inheritance and the offspring will have the same haplotype of the mitochondrial gene as the mother. Sea turtles are highly loyal to their birthplace and female adults return to their hatchling beach to lay their eggs. Thus, turtles from the same birthplace share unique haplotypes of the mitochondrial gene (*Bowen et al., 1992*; *Nishizawa et al., 2011*). Therefore, the mitochondrial gene can be used to identify the

birthplace of sea turtles, as well as the mother-child relationship between a baby turtle and an adult female. The combination of microsatellite and mitochondrial DNA markers could be used to rapidly exclude the parentage relationship and also trace the source of illegal trade or capture in forensic enforcement for the authenticity of artificially bred animals.

In 2019, a turtle farm in southern China claimed that six captive adult green turtles (four females and two males) had successfully been reproduced and more than 400 baby turtles hatched between 2015 and 2017. Although the farm has the breeding license, the success of artificial reproduction could not be verified, since it is not possible to provide evidence of sea turtle breeding, mating, and egg laying. To verify this, we collected skin samples from 19 adult turtles, including the above six potential parents, and 16 juvenile turtles randomly selected. The mitochondrial D-loop gene and two microsatellite loci were jointly sequenced and analyzed to detect whether a parent–offspring relationship existed between adult and juvenile turtles and to infer the birthplaces of these turtles.

## MATERIALS & METHODS

### Sample collection and DNA extraction

Skin samples of sea turtles were collected form their hind limbs and disinfected with iodophor (LIRCON, Shandong, China) after sampling. The sex of adult individuals was identified according to the length of the tail (*Zhang, Li & Wang, 1995*). All samples were stored in 95% alcohol between 15–25 °C. The sample collection in this work has been approved by the Chinese government, and this work was conducted in strict accordance with the guidelines of the Animal Research Ethics Committee of Hainan Provincial Education Centre for Ecology and Environment, Hainan Normal University (HNECEE-2012-005).

DNA was extracted using a Tiangen Blood/Cell/Tissue Genomic DNA Extraction Kit (TIANamp Genomic DNA Kit, Beijing, China), following the manufacturer's protocol. DNA concentration and quality were tested using a NanoDrop™ One/OneC spectrophotometer (Thermo Fisher, MA, USA).

### Amplification and detection of gene loci

The primers LCM15382 (5′-GCTTAACCCTAAAGCATTGG-3′) and H950g (5′-AGTCTCGGATTTAGGGGTTTG-3′) of the mitochondrial D-loop were used for this study, and the length of the amplified product was 770 bp (*Yang, 2015*; *Wei, 2016*).

The two microsatellite loci B103 and D1 are markers previously developed for green turtles, and these loci have high genetic diversity and were sufficient to ensure accuracy of the discrimination results (Table S1; *Dutton & Frey, 2010*). Using an online tool (http://www.primer-dimer.com/), we detected that these two pairs of primers did not dimerize; therefore, different fluorescent dye labels (TAMRA and FAM) were used, and the two pairs of primers were separately subjected to PCR and multiplex PCR to obtain two independent replicates.

The PCR amplification mixture (50 μl) contained 2 × Taq mix (RN03001S, MonAmp™), 25 μl; template DNA, 2 μl; 10 μM primer dilutions, each 2 μl, amonng multiplex PCR contains two10 μM primer dilutions (B103 and D1), 2 μl each;

supplemented with ultrapure water to 50 µl. PCR conditions were as follows: pre-denaturing at 94 °C for 3 min; followed by 35 cycles of denaturing at 94 °C for 40 s, annealing at 55 °C for 40 s, extension at 72 °C for 30 s; a final extension at 72 °C for 4 min; and storage at 4 °C. To prevent fluorescence primers from quenching, the samples were protected from light. Precisely, 5 µl of PCR product was loaded, and a 1% agarose gel was used for electrophoresis at 120 V/cm for 20 min, with Marker DL2000 (TIANamp, Beijing, China) as a reference. After the electrophoresis, the agarose gel was placed in a Molecular Imager® ChemiDoc™ XRS+ Imaging System (BIO-RAD, MA, USA) to capture images. PCR products were submitted to Guangzhou Qingke Technology Co., Ltd. for the first-generation sequencing of D-loop gene fragments and microsatellite loci typing. Amplicons were purified and sequenced in both directions using BigDye on an ABI 3730XL sequencing system (Applied Biosystems, Foster City, CA, USA) (*Gaillard et al., 2020*). And microsatellite allele sizes were estimated in 2 µL of diluted amplified DNA, 0.5 µL of GeneScanTM 500 Liz Size standard (Applied Biosystems) and 12.5 µL of deionized formamide on an ABI 3000 DNA Analyzer (Applied Biosystems), and alleles assigned using GeneMapper® software (version 3.7, Applied Biosystems) and GeneScan 500 ROX fluorescent size standard (Applied Biosystems) (*Barbanti et al., 2019*).

## Parentage testing

In this study, we adopted the direct exclusion method of forensic science to distinguish the authenticity of artificially bred individuals as follows: (1) If the adult female turtle did not carry the same haplotype as the juvenile turtle on the mitochondrial D-loop segment, it was excluded as the parent of the juvenile individual; (2) if an adult male or female turtle did not carry at least one matching allele in each of the microsatellite loci as the juvenile turtle, it was excluded as the parent of the juvenile individual; (3) adult individuals were paired using the microsatellite genotypes after the first two steps of screening. If the combination matched the juvenile individuals' microsatellite genotypes, then the corresponding male and female individuals were considered as potential parents.

## Mixed-stock analysis

To determine stock contributions for the all turtles in the present study, we used a Bayesian MSA approach, as described in the program Bayes (*Pella & Masuda, 2001*). We ran separate MSA analyses each for the juvenile and adult datasets. We used the same haplotype frequencies as that of Australasian nesting rookeries, according to *Gaillard et al. (2020)*. We used both uniform priors (UP) and an informative priors (IP) analysis, as reported by *Dethmers et al. (2010)* to determine stock contributions. For each analysis, 30 chains were run with 100,000 MCMC for every chain. For each chain in the UP analysis, a different stock was given 95% contribution with a burn-in set of 50,000. For each chain in the IP analysis, each stock's contribution was based on its estimated population size. The Gelman and Rubin shrink factor was used to determine convergence for each chain (*Pella & Masuda, 2001*). Stock contributions with a shrink factor of >1.2 after 100,000 MCMCs were considered invalid.

## RESULTS

### Sample information and DNA extraction

The dorsal carapace curve length of male individuals was 79.0–88.0 cm and the curve width was 69.8–81.0 cm. The dorsal carapace curve length of female individuals was 70.0–100.0 cm and the curve width was 64.0–87.9 cm. The dorsal carapace curve length of juvenile individuals was 32.3–57.0 cm and the curve width was 27.5–50.0 cm; they weighed 7.75–22.15 kg. The quality of DNA extracted from the samples was high, and the DNA concentration was above 4.8 ng/µl (Table S2).

### Amplification and detection results of gene loci

PCR amplification of mitochondrial D-loop gene fragments was successful, and sequencing results were satisfactory. And PCR amplification of the two microsatellites, D1 and B103, was also successful and the type scanning results were good.

The mitochondrial D-loop fragment had nine haplotypes in 19 adults and five haplotypes in 16 juvenile individuals. Microsatellite loci D1 had 11 allele lengths in 19 adults, and nine allele lengths in 16 juvenile individuals. Loci B103 had seven allele lengths in 19 adults and nine allele lengths in 16 juvenile individuals (Table S3).

### Parentage test results

By testing the parentage of two microsatellite loci, we found that only two of the 16 juvenile turtles had a possible parent combination. However, the D-loop genotype of the suspected female parent did not match that of the juvenile turtles. As a result, none of the 16 juvenile turtles had a matching parent combination of the 19 adult turtles (Table 1). Therefore, there was no parentage relationship between the 19 adult and 16 juvenile green turtles at the farm.

### Mixed-stock analysis

Among the 19 adult and 16 juvenile green turtles, 11 mitochondrial haplotypes were found. The CmP57.1 haplotype had the highest frequency, accounting for 34.3%, followed by CmP20.1, CmP19.1, and CmP49.3, accounting for 11.4%. In addition, a new haplotype type was found (Table S4).

Both the IP and UP MSA analyses suggested that the majority of stock contribution comes from the Sulu Sea for all datasets (IP 66.54%; UP 66.45% juveniles; IP 49.15%; UP 48.37% adults, Tables S5 and S6). Moderate stock contributions were shown to come from the Marshall Islands for adults but not juveniles (IP 22.10%; UP 21.22% adults; IP 0.21%; UP 0.21% juveniles), and moderate contributions came from the Paracel Islands for juveniles, but to a lesser extent for adults (IP 19.79%; UP 19.92% juveniles; IP 3.47%; UP 3.45% adults). Juveniles and adults had similar contributions from Wanan Island Taiwan (IP 4.97%; UP 4.87% juveniles; IP 4.90%; UP 5.62% adults). However, the 95% CI for all stock contributions, except for the Sulu Sea, included zero suggesting these are more of a general estimate than an absolute one (*Joseph et al., 2014*; *Jensen, Pilcher & Fitzsimmons, 2016*). Nonetheless, ~88% of stock contributions can be attributed to the Sulu Sea, Marshall Islands, Paracel Islands and Wanan Island. All shrink factors for stock contributions were less than 1.2.

Zhang et al. (2021), *PeerJ*, DOI 10.7717/peerj.12410

**Table 1  Parentage identification results using the direct exclusion method.**

| Numbering | Females with the same D-loop genotype | Adult individuals screened by microsatellite loci D1 and B103 | | | Identification results |
|---|---|---|---|---|---|
| | | **Males** | **Females** | **Suspected parent combination** | |
| B151 | A03(F)/A04(F)/A22(F) | \ | \ | \ | Unmatched parent combination |
| B152 | A03(F)/A04(F)/A22(F) | \ | A20(F)/A14(F)/A04(F) | \ | Unmatched parent combination |
| B153 | A03(F)/A04(F)/A22(F) | \ | \ | \ | Unmatched parent combination |
| B154 | \ | \ | \ | \ | Unmatched parent combination |
| B155 | \ | \ | A21(F) | \ | Unmatched parent combination |
| B161 | \ | \ | A13(F) | \ | Unmatched parent combination |
| B162 | A03(F)/A04(F)/A22(F) | A09(M) | A21(F)/A14(F)/ A07(F) | \ | Unmatched parent combination |
| B163 | A03(F)/A04(F)/A22(F) | A01(M)/A11(M) | \ | \ | Unmatched parent combination |
| B164 | \ | A23(M) | A10(F)/A22(F)/A16(F)/A02(F) | A23(M) × A10(F)/A02(F) | Unmatched parent combination[*] |
| B165 | \ | \ | A16(F)/A02(F)/A10(F) | \ | Unmatched parent combination |
| B166 | A06(F) | \ | \ | \ | Unmatched parent combination |
| B167 | A07(F) | A23(M) | A16(F)/A22(F) | \ | Unmatched parent combination |
| B171 | A06(F) | \ | \ | \ | Unmatched parent combination |
| B172 | A07(F) | A01(M)/A11(M) | A06(F) | A11(M) × A06(F) | Unmatched parent combination[*] |
| B173 | A07(F) | \ | A16(F)/A02(F) | \ | Unmatched parent combination |
| B174 | A03(F)/A04(F)/A22(F) | A01(M)/A09(M) | \ | \ | Unmatched parent combination |

**Notes.**
[*]Although there is a suspected parental combination at the microsatellite locus, the D-loop genotype of the suspected parent does not match.

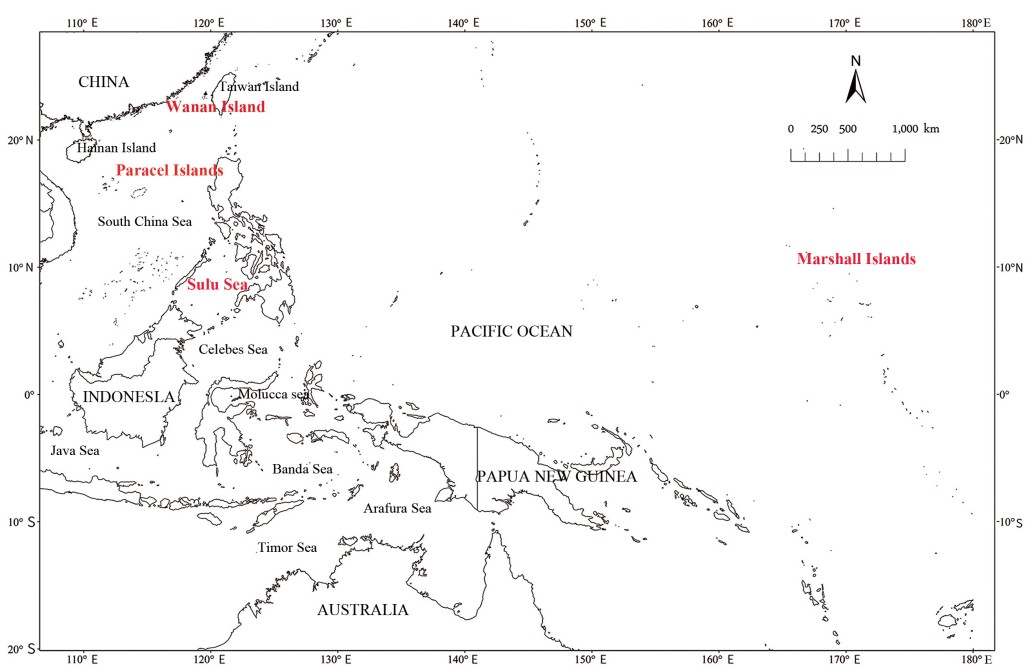

**Figure 1** Map of sea turtles birthplaces, mainly the Sulu Sea, the Marshall Islands, as well as Paracel Islands and Wanan Island in the South China Sea.

## DISCUSSION

Although this turtle farm claimed that the six adult green turtles bred all the 400 juvenile turtles in captivity, no parentage relationship could be identified between the randomly selected 16 juvenile turtles and the six adult green turtles, or even to the other 13 adults. Thus, it is evident that this farm is dishonest, is illegally farming and trading wild sea turtles, and marketing them as artificially bred turtles. The more surprising fact is that the farm has a breeding license issued by the local government. The legal license facilitates the laundering of illegal business in this case, similar to other cases found in China (*Lin et al., 2021*). These sea turtles in captivity may have various illegal overseas sources, their birthplace was mainly the Sulu Sea, followed by the Marshall Islands, as well as Paracel Islands and Wanan Island in the South China Sea (Fig. 1). Thus, there is a great possibility that this farm purchased wild turtle eggs or hatchlings from the above birthplaces through illegal trade chains. Further investigation is warranted to uncover the detailed source and illegal trade chain by the law enforcement staff.

The huge demand for sea turtles has made China a major market and destination of illegal trade for a long time, and most traded sea turtles are mainly from fishermen who captured turtles directly or purchased from dealers in southeast Asian countries (*Lam et al., 2011*; *Gomez & Krishnasamy, 2019*; *Lin et al., 2021*). Wild sea turtle populations in the South China Sea and the Coral Triangle area suffer the most, which has been revealed by mix-stock analysis of sea turtle samples from the Hainan Islands, China (*Gaillard et al., 2020*), Brunei Bay and Mantanani, Malaysia (*Joseph et al., 2014*; *Joseph et al., 2016*). The

results of the present study are consistent with their findings and provide further evidence on the gravity of illegal turtle fishing and trade in these areas. Our data show juveniles and adults are generally collected from the Sulu Sea, but that hatchlings are more heavily harvested from the Paracel Islands than in adults. However, adults are being collected from distant rookeries as the Marshall Islands made up over 20% of their stock contribution. It could be that adult turtles are migrating to the feeding grounds in the Sulu Sea and are being collected there, however, previous analysis of foraging grounds in the Sulu Sea did not find stock contributions from the Marshall Islands (*Jensen, Pilcher & Fitzsimmons, 2016*; *Joseph et al., 2016*) suggesting these animals were collected from a rookery not found in the Sulu Sea. Therefore, regarding the current sea turtle illegal trade, stricter and more efficient law enforcement is crucial (*Lin et al., 2021*), and simultaneous monitoring of market trends and trade routes must occur (*Lam et al., 2011*), multinational cooperation is also required to properly protect green turtles in the South China Sea.

To the best of our knowledge, the present study is the first attempt to use nuclear microsatellite gene combined with mitochondrial gene markers for authenticity testing of artificially bred green turtles in forensic enforcement. The trinucleotide (B103) and tetranucleotide (D1) microsatellite loci used in this study have high genetic diversity and are sufficient to ensure the accuracy of the discrimination results, while the D-loop fragments in mitochondrial DNA are mostly used to identify the origin of sea turtles (*Gaos et al., 2017*; *Hill et al., 2018*). Their combination is reliable to rapidly distinguish the authenticity of artificial breeding and also accurately trace the source of the illegal trade or capture of sea turtles. Meanwhile, the testing is fast and cheap, as the samples can be tested within 48 h and each sample costs about 200 RMB on average. However, due to limited conditions, only 16 juvenile and 19 adult turtles were sampled in this study, which could be successfully identified using two microsatellite and one mitochondrial loci. But for testing a larger number of samples, it is recommended to increase the number of microsatellite loci and use the multiplex PCR method adopted in this study to ensure the accuracy of detection. Microsatellite loci, such as A6, B123, C102, Cm3, Cm58, etc. used by *Wright et al. (2012)*, could be selected to satisfy the identification of more samples. As such, we believe that this method can serve as a good reference for identifying artificially bred green turtles in farms or as part of illegal trade and help obtain reliable evidence of turtle origins in future law enforcement.

## CONCLUSIONS AND CONSERVATION IMPLICATIONS

The turtle farm investigated in the present study is illegally trading wild sea turtles and marketing them as artificially bred turtles. This method provides reliable evidence for this conclusion and also helps reveal the possible illegal trade chain. As the number of facilities for artificial breeding of sea turtles is expected to gradually increase with the encouragement of artificial breeding of sea turtles, this method could help distinguish the authenticity of artificial breeding in law enforcement and assess its impact on sea turtle populations in future. Moreover, international cooperation between China and Southeast Asian countries should be strengthened to combat the illegal trade of sea turtles and protect their wild populations in the South China Sea and Coral Triangle areas.

## ACKNOWLEDGEMENTS

We thank the reviewers for their helpful and insightful comments to improve our manuscript. We are also grateful to L. Kong, R. Li, and D.H. Cheng for their assistance with DNA extraction.

### Funding

This work was supported by the National Natural Science Foundation of China (31960101, 31772486), Hainan Natural Science Foundation (319MS048). The funders had no role in study design, data collection and analysis, decision to publish, or preparation of the manuscript.

### Grant Disclosures

The following grant information was disclosed by the authors:
The National Natural Science Foundation of China: 31960101, 31772486.
Hainan Natural Science Foundation:  319MS048.

### Competing Interests

The authors declare there are no competing interests.

### Author Contributions

- Ting Zhang and Liu Lin conceived and designed the experiments, performed the experiments, analyzed the data, prepared figures and/or tables, authored or reviewed drafts of the paper, and approved the final draft.
- Daniel Gaillard analyzed the data, prepared figures and/or tables, authored or reviewed drafts of the paper, and approved the final draft.
- Fang Chen, Shannan Wu and Zhao Wang performed the experiments, prepared figures and/or tables, and approved the final draft.
- Huaiqing Chen conceived and designed the experiments, analyzed the data, prepared figures and/or tables, authored or reviewed drafts of the paper, and approved the final draft.
- Meimei Li performed the experiments, authored or reviewed drafts of the paper, and approved the final draft.
- Haitao Shi conceived and designed the experiments, performed the experiments, authored or reviewed drafts of the paper, and approved the final draft.

### Data Availability

    The raw measurements are available in the Supplemental Files.

### Supplemental Information

Supplemental information for this article can be found online at http://dx.doi.org/10.7717/peerj.12410#supplemental-information.

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
