# Peer review of "Rapid authenticity testing of artificially bred green turtles (Chelonia mydas) using microsatellite and mitochondrial DNA markers"

_PeerJ, doi:10.7717/peerj.12410_

## Round 0.1 · original submission · Minor Revisions

Both reviewers agree that your paper needs some modifications before it can be published. Please follow their indications (especially those of Rev:1) to make the suggested changes or justify why not to make them and send us the paper for their consideration.

Reviewer 1 ·

Basic reporting

The raw mitochondrial haplotype data and microsatellite allele data are provided.

A figure illustrating the possible Australasian sources of these turtles would be helpful, rather than having to refer to other manuscripts for context.

I would recommend restructuring some of the information currently contained in the tables/supplemental tables.

Table 1 is unnecessary as it stands. I would recommend naming these loci and referring to the original development note and simply stated the dye used in the methods text. It might be more helpful to readers to treat this as a supplemental table and include more specifics of primer concentrations used in the multiplex PCR analysis.

The data presented in Table 3 might be better presented in a supplemental table with haplotypes as rows and captive adults and juveniles as separate columns, along with all of the rookery and foraging ground data available for this region. It should be fairly easy to append these data to an existing excel file from recent analyses. Alternatively, you could retain Table 3 but have separate columns to summarize the adults and juvenile haplotype counts from the present study and exclude the area of origin and frequency portion but refer to a supplemental table with these data.

Table 4 might be best treated as a supplemental table as well.

Experimental design

More details on the mitochondrial sequencing (or at least a reference to a previous study that used the same approach) are needed.

More details on the multiplex genotyping would also be helpful for authors interested in replicating this study.

Line 94 in materials and methods. I don't think skin biopsies qualify as noninvasive. I would recommend removing this descriptor.

For the mixed stock analysis, were only the 16 juveniles used or a mixture of adults and juveniles? It's not clear from the methods or results, given some haplotype overlap between groups. Please clarify. I would recommend only the juveniles be used or that they be treated separately.

There are several references to "paternity" testing when I think the authors are analyzing both parents. I would recommend referring to these as "parentage" analyses. Paternity specifically refers to potential sires/males.

Line 208/209: I would recommend referring to these as "trinucleotide" and "tetranucleotide" loci to follow standard nomenclature.

Validity of the findings

The loci used were sufficient in this case to discriminate between candidate parents and the randomly selected juveniles. However, as noted by authors, only two loci may not be enough in many other circumstances. I would recommend running many more loci to increase confidence that exclusion will be certain and also to help clarify potential relationships among juveniles of unknown origins.

There are existing resources for multiplex genotyping of green turtle microsatellite loci. The authors don't appear to have considered the available data. Previous studies have successfully employed mutiplex PCR reactions to provide data on multiple paternity and relatedness analysis (Wright et al., 2012, 13 well behaved loci in 2 multiplexes) and Barbanti et al. 2019, a different set of 13 loci in 2 multiplexes). Wright et al. 2012 previously demonstrated that D105 used in the present study had amplification/scoring issues and was an unreliable marker. I would recommend making use of these existing multiplexes that contain additional loci and recommending their use in the discussion section.

The authors successfully demonstrate the primary objective of excluding the adult farm turtles as the parents of the juveniles. However, this represents the bare minimum of meaningful treatment of these samples, particularly given that only 16 of several hundred juveniles were analyzed. I appreciate that resources are limited, but relatedness analysis of at least the randomly selected juveniles already using either set of the 2 multiplexes above could provide important additional context on their potential source(s). Are the juveniles that share haplotypes siblings or half siblings or more likely unrelated? If this is beyond the scope of the present study, I would recommend running additional loci and additional samples in the future.

Additional comments

Rather than referring to the mitochondrial sequences and microsatellite loci analyzed as "genes" I would simply call them sequences/haplotypes or microsatellite loci.

Reviewer 2 ·

Basic reporting

There are some key papers that can add crucial context to this study.

Shi et al. 2007. Farming endangered turtles to extinction in China. Conservation Biology 21(1):5-6.

This study discusses a related problem for farming freshwater turtles in China. Given the parallels with this study the authors should include the direct comparison. One of the key points is that farms are often the biggest purchasers of wild turtles.

As part of this, the bushmeat phenomenon and false localities of Chinese turtles led to a series of studies that talked about false claims of provenance.

Parham, J.F., and H. Shi. 2001. The discovery of Mauremys iversoni-like turtles at a turtle farm in Hainan Province, China: The counterfeit golden coin. Asiatic Herpetological Research 9:71-77.

Parham et al. 2001. New Chinese turtles: Endangered or invalid? A reassessment of two species using mitochondrial DNA, allozyme electrophoresis, and known locality specimens. Animal Conservation 4(4):357-367.

Li et al. 2021. Sea turtle demand in China threatens the survival of wild populations. iScience 24(6):102517.

This recent study on the plight of captive sea turtles in China discusses faux-farming situations (aka ranching) and should be cited as it directly relates to the current study (both specifically and in general).
* * *
Line 22: “which further aggravates the already endangered species” is awkward as it sounds like it makes the species angry. I suggest a different phrasing “which adds another threat to an already endangered species”

Experimental design

no comment

Validity of the findings

no comment

Additional comments

I think this is a really simple and straightforward, but also elegant and important, study. I provide some citations that can help better-frame the study,

---

## Round 0.2 · Minor Revisions

The revised version need some explanation on the mixed stock analysis as it was suggested by the reviewer. Please take your time to analyze the reviewer suggestion or explain your reason for do not doing it. I will be waiting it.

Reviewer 1 ·

Basic reporting

This version of the manuscript looks much better.

Experimental design

The revised version rectified most potential issues with methodology and analyses. The one glaring remaining issue is with the mixed stock analyses. Despite the small sample sizes, the haplotype frequencies between the adults and juveniles looked like they might be differentiated to me. I ran the frequencies through Arlequin. The FST comparison was not significant, but the exact test did indicate significant differentiation (p = 0.025). It is inappropriate to combine these groups for mixed stock analyses given this signal of differentiation and the nuclear data that exclude them as parents. Analyzing small sample sizes through MSA will certainly lead to broader credible intervals, which rightfully indicate greater uncertainty given potential for sampling error with smaller data sets. But that would be better than combining disparate samples and potentially getting illegitimate signals about their sources. I would recommend splitting these into 2 groups and rerunning the analyses. It should only take an hour to create 2 separate DEC files and run them both through BAYES with all of the existing base and control files you already have.

Validity of the findings

No comment.

---

## Round 0.3 · accepted · Accept

Dear Authors,

Thank you for addressing the minor revisions.